# Salivary Transmembrane Mucins of the MUC1 Family (CA 15-3, CA 27.29, MCA) in Breast Cancer: The Effect of Human Epidermal Growth Factor Receptor 2 (HER2)

**DOI:** 10.3390/cancers16203461

**Published:** 2024-10-12

**Authors:** Elena I. Dyachenko, Lyudmila V. Bel’skaya

**Affiliations:** Biochemistry Research Laboratory, Omsk State Pedagogical University, 644099 Omsk, Russia; dyachenko.ea@gkpc.buzoo.ru

**Keywords:** saliva, breast cancer, MUC1 family, CA 15-3, CA 27.29, MCA, cytokines, amino acids

## Abstract

**Simple Summary:**

It was shown for the first time that the salivary levels of the MUC1 family, including CA 15-3, CA 27.29, and MCA, were significantly dependent on the expression of human epidermal growth factor receptor 2 (HER2) in breast cancer patients. In the presence of HER2 expression, mucin levels decreased with a simultaneous decrease in the levels of free estrogen and progesterone and an increase in the content of pro-inflammatory cytokines and free amino acids, which was consistent with the aggressive growth pattern and high invasiveness of HER2-positive breast cancer subtypes.

**Abstract:**

The MUC1 family of transmembrane glycoproteins (CA 15-3, CA 27.29, MCA) is aberrantly expressed among patients with breast cancer. **Objectives**: to measure the level of degradation products of MUC1, including CA 15-3, CA 27.29, and MCA, in the saliva of breast cancer patients and to describe the biochemical processes that influence their expression and the regulation of their biological functions. **Methods**: The case–control study included three groups (breast cancer, fibroadenomas, and healthy controls). All study participants provided saliva samples strictly before starting treatment. The levels of MUC1, including CA 15-3, CA 27.29, and MCA, free progesterone and estradiol, cytokines (MCP-1, VEGF, TNF-α, IL-1β, IL-2, IL-4, IL-6, IL-8, IL-10, IL-18), and amino acids (Asp, Gln, Gly, His, Leu + Ile, Orn, Phe, Pro, Tyr) were determined. **Results**: It was shown that the levels of the MUC1 family in the saliva of patients with HER2-positive breast cancer were significantly lower compared to the control group. The level of pro-inflammatory cytokines and the level of free estradiol affected the expression of MUC1. We obtained a reliable relationship between the aggressive nature of tumor growth, an increased level of pro-inflammatory cytokines, a low level of free estradiol, and the suppressed expression of salivary MUC1. **Conclusions**: Among patients with aggressive breast cancer, a high level of pro-inflammatory cytokines, and a low level of free estradiol, there was an inhibition of the expression of pathologically unchanged glycoprotein MUC1 in saliva.

## 1. Introduction

Mucins have multiple roles in the human body, both in healthy and pathological conditions, including cancer [1,2,3]. This is primarily because of the structural features that determine their functional activity. Epithelial cells (goblet and luminal [4]) and hematopoietic cells can express mucins [5]. All mucins are referred to as the family of large glycoproteins and are the main component of mucus. To date, mucins are usually classified into two major large types: membrane-bound and secreted (soluble) mucins. The membrane-bound mucins include MUC1, MUC3A, MUC3B, MUC4, MUC12, MUC13, MUC14, MUC15, MUC16, MUC17, MUC20, MUC21, and MUC22. The secreted type of mucin is divided into two subtypes, gel-forming mucins (MUC2, MUC5AC, MUC5AB, MUC6, and MUC19) and non-gel-forming mucins (MUC7, MUC8, and MUC9) [6]. Membrane-bound mucins provide intercellular communication, participate in immune reactivity and cell cycle regulation, and are located on the apical and lateral surfaces of cells [7,8,9]. Secreted mucins perform a protective role by forming a gel-like substance [6]. Every mucin creates its own family based on a variety of epitopes in its extracellular domain. The following classification would be massive and is not our main goal. Thus, we decided to stop only on the description of the MUC1 family because it is the object of our research.

MUC1, also known as an episalin, is located on chromosome 1q22 and is the first mucin whose gene was cloned and well characterized [10]. It belongs to the transmembrane mucins and consists of an extracellular domain (N-glycosylated domain) containing a multivariable number of tandem repeats of 20 amino acids, a sea urchin sperm protein enterokinase and agrin (SEA) domain characteristic only of MUC1, a transmembrane domain, and a cytoplasmic domain (C-glycosylated domain) consisting of 72 amino acids [11,12]. The molecular weight of the mature glycosylated protein is 250–500 kDa [13]. The variable number of tandem repeats (VNTR) is highly conserved, containing mainly Pro, Thr, and Ser residues [14,15,16,17,18]. Glycosylation occurs at the pair of tandem repeats Ser and Thr [6]. Each such pair has five glycosylation sites. Between the tandem repeat sites, there is an immunodominant region containing epitopes [19]. MUC1 is a highly immunogenic molecule, so any change occurring in the human body that affects the homeostatic balance will lead to a change in metabolism in the cell. As a result, mucins will have a changed composition and length of the N-glycosylated domain (the domain will be shorter with infrequent glycosylation), exposing the immunogenic epitopes that are perceived by the immune system as antigens. Altered cellular metabolism in pathology also induces the autolytic cleavage of the N-glycosylated domain [20], exposing SEA and the transmembrane domain, which in this case will serve as a receptor for protein kinase C delta, glycogen synthase kinase 3β and epidermal growth factor [21,22,23]. The formation of mucins directly depends on glycosyltransferases and sialyltransferases. It has been shown that the position and number of these enzymes in the Golgi apparatus directly affect the length and expression of Cor1 (Galβ1,3 GalNAc) and Cor2 (GlcNAcTβ1,6 Galβ1,3 GalNAc) glycans and also change their activity depending on the type of cancer [24,25].

The MUC1 family includes antigens CA 15-3 (M12), CA 27.29 (M20), and MCA (M22), which have similar epitopes to MUC1 [26,27,28,29,30,31,32,33]. The commonality among these antigens is that they are used to diagnose breast cancer.

CA 15-3 belongs to the soluble part of MUC1; the gene is localized on 7q22. The mucin-like domain consists of 28 repeating amino acids, mainly Pro, Thr, and Ser. Its non-mucin-like domain consists of 150 amino acids, flanked by Cys-EGF-like domains [34]. The protein structure also includes a transmembrane domain and a cytoplasmic long domain consisting of 75 amino acids and with a biologically important sequence, Tyr-Asn-Asn-Phe. This amino acid sequence is recognized by proteins with SH2 groups, which trigger intracellular signaling pathways [35]. The protein consists of linked C- and N-glycosylated terminal subunits and a heavily glycosylated extracellular domain. This protein is present in many epithelial cells [36]. Today, a measurement of the level of the oncogene CA 15-3 is actively used in routine laboratory practice during the examination of patients with breast cancer to confirm the diagnosis, monitor patients dynamically, and respond to therapeutic intervention [37,38]. Free CA 15-3 is used as a prognosis and response to therapy [39,40], while measurement is not recommended for monitoring asymptomatic early breast cancer [41].

CA 27.29 is a glycoform of MUC1. It is localized on chromosome 3q29 [26]. It has high specificity but low sensitivity compared to CA 15-3. Different levels of CA 27.29 and CA 15-3 may be connected with different factors: the test system and its kit; patient sample; and patient age (it was found that CA27.29 and CA 15-3 levels are higher in postmenopausal people than in premenopausal people) [42]. CA 27.29 has been shown to be significantly higher in elderly people and women with postmenopausal status [43,44,45]. This is explained by a decrease in the sialylation caused by aging, resulting in the exposure of immunogenic epitopes of MUC1, which results in increased values of both MUC1 itself and its glycoform CA 27.29 [46].

MCA (Mucin-Like Carcinoma Antigen) belongs to the mucin family and is localized on chromosome 6p21.3 [47]. After analyzing tandem repeats of 20 amino acids in the MUC1 protein, glycosylation sites, and the epitope structure, it was established that MCA belongs to MUC1 [48,49,50]. Using MCA as the only marker of breast cancer is not advisable, since it can increase in pregnant women after the 4th month of pregnancy, with benign neoplasms and advanced stages of breast cancer [51]. At the same time, the combination of MCA with CA 27.29 and CA 15-3 increases the sensitivity of breast cancer diagnosis [52,53]. MCA testing alone has been useful for predicting disease recurrence, screening for metastases, and during chemotherapy [54].

The FDA [55] and the American Society of Clinical Oncologists [46] have approved a combination of tumor markers CA 15-3, CA 27.29, and MCA for use in laboratory diagnostics of breast cancer [56]. These markers are glycoproteins and belong to the general MUC1 family. Tumor markers CA 15-3, CA 27.29, and MCA are antigens and have unique tandem repeats containing highly immunodominant epitopes in the structure of the MUC1 molecule. This combination of mucins increases the sensitivity of the test system due to MCA [57] and specificity due to CA 27.29 [46]. However, serum/plasma is used to determine mucins, while the use of saliva for these purposes is promising.

The aim of this study is to measure the level of degradation products of MUC1, including CA 15-3, CA 27.29, and MCA, in the saliva of breast cancer patients and to describe the biochemical processes that influence their expression and the regulation of their biological functions.

## 2. Materials and Methods

### 2.1. Study Design

The study included 381 participants, including 230 patients with breast cancer (age 60.0 [47.8; 66.8] years), 92 patients with fibroadenomas (age 44.7 [38.8; 57.0] years), and 59 healthy controls (age 44.9 [36.1; 52.7] years). Cytokine determination was performed in all participants (n = 381). Then, 110 patients with breast cancer were selected for mucin determination in such a way that the subgroups contained an equal number of patients with each molecular biological subtype of breast cancer (n = 110; 22 people in a subgroup). From those patients in whom cytokines and mucins were determined, samples were selected for amino acid determination. Since some samples were insufficient in volume for amino acid analysis, samples were selected from those in whom only cytokines were determined. In this regard, the sample sizes differed slightly, as did the number of patients in the subgroups. However, the majority of samples were used to conduct all three studies in parallel. A detailed description of the study group is given in Table 1. cancers-16-03461-t001_Table 1Table 1Characteristics of the study groups.FeatureMucinsCytokinesAmino Acids**Control group**n = 30n = 59n = 25**Fibroadenomas**n = 26n = 92n = 24**Breast cancer**n = 110n = 230n = 116Clinical stageStage I + II62 (56.3%)171 (74.3%) 80 (69.0%)Stage III + IV48 (43.7%)59 (25.7%)36 (31.0%)Degree of differentiation (G)G I + II46 (41.8%)104 (45.2%)74 (63.8%)G III47 (42.7%)92 (40.0%)42 (36.2%)Unknown17 (15.5%)34 (14.8%)-HER2 statusHER2(−)66 (60%)159 (69.1%)88 (75.9%)HER2(+)44 (40%)62 (27.0%)28 (24.1%)Unknown-9 (3.9%)-Estrogen (ER) statusER (−)47 (42.7%)73 (31.7%)26 (22.4%)ER (+)63 (57.3%)149 (64.8%)90 (77.6%)Unknown-8 (3.5%)-Progesterone (PR) statusPR (−)59 (53.6%)103 (44.8%)46 (39.7%)PR (+)51 (46.4%)119 (51.7%)70 (60.3%)Unknown
8 (3.5%)-Proliferative activity index Ki-67Low (<20%)34 (30.9%)117 (50.9%)59 (50.9%)High (>20%)76 (69.1%)93 (40.4%)57 (49.1%)Unknown-20 (8.7%)-Molecular biological subtypeLuminal A22 (20.0%)61 (26.5%)40 (34.5%)Luminal B (HER2−)22 (20.0%)57 (24.9%)35 (30.2%)Luminal B (HER2+)22 (20.0%)33 (14.3%)15 (12.9%)Non-luminal22 (20.0%)30 (13.0%)12 (10.3%)Triple-negative22 (20.0%)41 (17.8%)14 (12.1%)Unknown-8 (3.5%)-

The study included female patients without age restrictions. The patients were hospitalized for planned surgical treatment or the first course of chemotherapy. After histological verification, we classified the patients into the corresponding subgroups (breast cancer or fibroadenomas). The inclusion criteria also included the absence of inflammatory processes, viral diseases, and oral sanitation. The patients of the control group were recruited from among volunteers at the blood transfusion station, in whom mammography revealed no breast pathologies. The studies were approved at a meeting of the ethics committee of the Omsk Region Clinical Oncology Dispensary on 21 July 2016, protocol No. 15 and Institutional Review Board No. 46-04/2.

### 2.2. Collection of Saliva Samples

Saliva samples were collected by spitting into sterile polypropylene centrifuge tubes (2–3 mL volume) between 8 and 10 a.m. on an empty stomach after the preliminary rinsing of the mouth with water. To precipitate cells and impurities, centrifugation was performed at 10,000× *g* for 10 min (CLb-16, Moscow, Russia). Then, 1 mL of the upper layer was collected, transferred to Eppendorf tubes, and stored at −80 °C until analysis.

### 2.3. Determination of Total Mucin and MUC1 in Saliva

The total mucin content in saliva was determined by the difference between the protein content in saliva and in the supernatant after the preliminary precipitation of mucin with a 20% acetic acid solution [58].

The content of MUC1, including CA 15-3 (catalog number K226), CA 27.29 (K227), and MCA (K228) in saliva was determined by solid-phase enzyme immunoassay using Hema kits (Moscow, Russia) on a Thermo Fisher Multiskan FC analyzer (Waltham, MA, USA). The aliquot volume in all cases was 100 μL. The analysis and calculation of the mucin level (U/mL) was carried out in accordance with the manufacturer’s instructions.

### 2.4. Determination of Estradiol and Progesterone in Saliva

Free estradiol (K208) and progesterone (K207) in saliva were determined by solid-phase enzyme immunoassay using Hema kits (Moscow, Russia). The aliquot volume was 25 μL; the analysis was performed in accordance with the manufacturer’s instructions on a Thermo Fisher Multiskan FC analyzer (Waltham, MA, USA).

### 2.5. Determination of Cytokines in Saliva

Cytokines, including TNF-α (catalog number A-8756), MCP-1 (A-8782), VEGF (A-8784), IL-1β (A-8766), IL-2 (A-8772), IL-4 (A-8754), IL-6 (A-8768), IL-8 (A-8762), IL-10 (A-8774), and IL-18 (A-8770), were determined in saliva using solid-phase enzyme-linked immunosorbent assay using Vector-Best kits (Novosibirsk, Russia) on a Thermo Fisher Multiskan FC analyzer (Waltham, MA, USA).

### 2.6. Determination of the Amino Acid Composition of Saliva

High-performance liquid chromatography was used to obtain the amino acid profile of saliva with mass spectrometric detection in the selected reaction monitoring mode (1260 Infinity II chromatograph (Agilent, Santa Clara, CA, USA), 6460 Triple Quad detector (Agilent, USA)). Samples were separated on an Agilent Zorbax Eclipse XDB-C18 2.1 × 100 mm column with a sorbent diameter of 1.8 μm (Agilent, USA). The internal standard method (alanine-d4) was used for the reverse calculation of the level. At least six samples of the Amino Acids kit (Jasem, İstanbul, Turkey) were used to construct the calibration scale. The automatic integration of chromatograms using the Quantitative Quant-my-way software (MassHunter Workstation Quantitative Analysis B.09.00, Agilent, Santa Clara, CA, USA) was used to analyze the results.

### 2.7. Determination of the Expression of the Receptors for Estrogen, Progesterone, HER2, and Ki-67

The determination of the expression levels of estrogen receptors (ER), progesterone receptors (PR), and HER2 was performed in accordance with the Allred assessment guidelines and ASCO/CAP recommendations [59,60]. Ki-67 expression was determined according to the manufacturer’s protocol with a cutoff value of 20% [61].

### 2.8. Statistical Analysis

Statistical analysis of the data was performed using the nonparametric method in the Statistica 13.3 EN program (StatSoft, Tulsa, OK, USA). The nature of the distribution and homogeneity of variances in the groups were preliminarily tested. According to the Shapiro–Wilk test, the content of all the parameters determined does not correspond to the normal distribution (*p* < 0.05). The conducted test for the homogeneity of variances in the groups (Bartlett’s test) allowed us to reject the hypothesis that the variances are homogeneous across groups (*p* < 0.0001). Therefore, nonparametric statistics were used to process the data. The results are presented as the median (Me) and interquartile range in the form of the 25th and 75th percentiles [LQ; UQ]. Differences were considered statistically significant at *p* ˂ 0.05.

## 3. Results

### 3.1. Changes in Salivary Level of CA 15-3, CA 27.29, MCA in Breast Cancer

In the first stage of our experiment, we determined the level of the total mucin in saliva. It was shown that in breast cancer, the level of mucin was 0.282 [0.103; 0.561] g/L, while in the saliva of the control group, the level was significantly higher at 0.560 [0.434; 0.727] g/L (*p* = 0.0003). The following stage of our study was a comparison of the content of different antigens of the MUC1 family in the saliva that are associated with breast cancer.

It was shown that the level of MUC1 CA 15-3, CA 27.29, and MCA in saliva also decreased (Figure 1), but this decrease was statistically insignificant (*p* = 0.3038, 0.2882 and 0.5439, respectively). It should be noted that the levels of mucins CA 27.29 and MCA in saliva in fibroadenomas, on the contrary, increased. For mucin CA 27.29, the increase in level in fibroadenomas was statistically significant (*p* = 0.0318).
Figure 1The levels of mucins MUC1 CA 15-3, CA 27.29, MCA in saliva in the control group, fibroadenomas, and breast cancer (U/mL). BC—breast cancer (n = 110); FA—fibroadenomas (n = 26); HC—healthy control (n = 30). *—differences with breast cancer are statistically significant, *p* < 0.05.
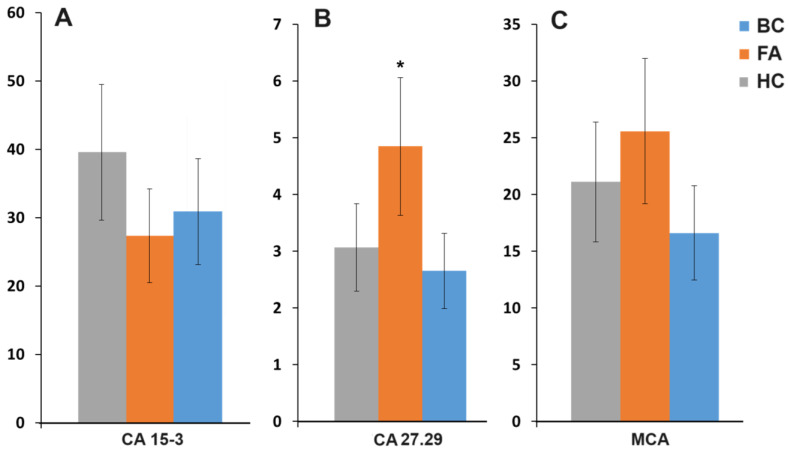


### 3.2. The Level of CA 15-3, CA 27.29, MCA with Different Expression of HER2 Depending on the Stage of the Disease (St), the Degree of Differentiation (G), and the Proliferative Activity Index Ki-67

There were no statistically significant differences in the level of MUC1 mucins depending on the stage, degree of differentiation, or proliferative activity index of breast cancer or in the expression status of estrogen and progesterone receptors (*p* ˃ 0.05). The only factor affecting the level of mucins CA 15-3, CA 27.29, and MCA in saliva was the expression of human epidermal growth factor receptor 2 (HER2) (Table 2). It was shown that the level of CA 15-3, CA 27.29, and MCA statistically significantly decreased in HER2-positive breast cancer (−50, −53, and −63% for CA 15-3, CA 27.29, MCA, respectively).cancers-16-03461-t002_Table 2Table 2The level of mucins CA 15-3, CA 27.29, and MCA in saliva in breast cancer, depending on the expression of HER2.Mucins, U/mLBreast Cancer, n = 110Healthy Control, n = 30 (Group 3)Fibroadenomas, n = 26 (Group 4)HER2(+), n = 44(Group 1)HER2(−), n = 66(Group 2)CA 15-319.8 [9.1; 78.3] *37.4 [18.5; 84.7]39.6 [21.4; 92.7]27.4 [15.1; 73.1]*p1-2* = 0.0327*p1-2* = 0.0327--CA 27.291.44 [1.05; 3.72]3.83 [1.91; 7.23]3.08 [2.11; 5.22]4.85 [2.03; 8.70]*p1-2* = 0.0005*p1-3* = 0.0018*p1-4* = 0.0002*p1-2* = 0.0005*p1-3* = 0.0018*p1-4* = 0.0002MCA7.73 [2.81; 33.30]18.00 [9.36; 56.8]21.10 [6.24; 75.9]25.58 [9.89; 91.0]*p1-2* = 0.0026*p1-3* = 0.0419*p1-4* < 0.0001*p1-2* = 0.0026*p1-3* = 0.0419*p1-4* < 0.0001Note. *—Here and further in Table 3, Table 4, Table 5 and Table 6, the values are given as the median (Me) and interquartile range in the form of the 25th and 75th percentiles [LQ; UQ].

It was found that the level of mucins MUC1 CA 15-3, CA 27.29, and MCA in saliva with different HER2 expression statuses changed differently (Table 3).cancers-16-03461-t003_Table 3Table 3The level of mucins CA 15-3, CA 27.29, and MCA, depending on the clinical stage (St), the level of cell differentiation (G), and the proliferative activity index Ki-67.Mucins, U/mLHER2(+), n = 44HER2(−), n = 66*p*-ValueClinical StageSt I-II, n = 25 (Group 1)St III-IV, n = 19 (Group 2)St I-II, n = 30 (Group 3)St III-IV, n = 36 (Group 4)
CA 15-322.56 [10.72; 51.94]18.58 [6.84; 175.52]30.11 [16.64; 84.79]37.40 [22.76; 79.22]-CA 27.291.54 [1.08; 2.64]1.32 [0.67; 5.04]3.65 [1.31; 7.47]4.09 [2.02; 7.14]*p1-3* = 0.0187*p1-HC* = 0.0018MCA7.34 [2.60; 19.87]8.33 [6.43; 67.49]50.32 [9.53; 80.73]17.40 [9.34; 22.26]*p1-3* = 0.0008*p1-HC* = 0.0193**Degree of differentiation**GI + II, n = 16 (Group 1)GIII, n = 25 (Group 2)GI + II, n = 30 (Group 3)GIII, n = 24 (Group 4)*p*-value
CA 15-321.77 [9.70; 57.74]17.10 [9.49; 83.65]36.15 [18.22; 84.69]30.67 [18.35; 88.28]-CA 27.291.32 [1.06; 3.86]2.25 [1.09; 3.89]4.33 [1.67; 9.13]3.40 [2.02; 7.14]*p1-3* = 0.0224*p1-HC* = 0.0104*p2-HC* = 0.0295MCA6.88 [1.96; 9.55]15.53 [3.37; 37.86]17.40 [12.60; 71.97]19.28 [8.94; 50.49]*p1-3* = 0.0005*p1-HC* = 0.0137**Proliferative activity index Ki-67**Low, n = 7 (Group 1)High, n = 37 (Group 2)Low, n = 27 (Group 3)High, n = 39 (Group 4)*p*-value
CA 15-313.93 [8.68; 63.54]22.56 [9.49; 83.65]36.94 [19.39; 75.83]37.86 [16.64; 89.90]-CA 27.291.33 [0.66; 2.64]1.54 [1.07; 3.89]2.93 [1.52; 6.72]4.35 [2.17; 7.47]*p2-4* = 0.0018*p1-HC*
_=_ 0.0359*p2-HC* = 0.0037MCA7.45 [2.19; 10.50]8.33 [3.35; 35.27]17.71 [12.60; 72.62]18.22 [9.23; 44.24]*p1-3* = 0.0100Note. HC—healthy control. *p*-values are given for a comparison of subgroups with each other; the subgroup number is indicated in the column heading. When comparing a subgroup with a healthy control, the HC is indicated in the *p*-value index. The corresponding values for the healthy control are given in Table 2.

It is interesting to note that CA 15-3 and CA 27.29 behaved in the same way at different stages of the disease. In the HER2(+) group at stage St I-II, the level was higher for CA 15-3 (+21.4%) and CA 27.29 (+16.7%) than at stage St III-IV, while with HER2(−), the level of CA 15-3 and CA 27.29 was higher at widespread stages of the disease (+24.2% and +12.1%, respectively). The nature of the change in the level of MCA was different. In the HER2(+) group, with an increase in stage, it increased (+13.5%), while in the HER2(−) group, the level of MCA, on the contrary, significantly decreased (−65.4%). Depending on the degree of differentiation, MCA had a unidirectional nature of change. It is noteworthy that in the HER2(+) and HER2(−) groups, the level of MCA was higher at GIII by +125.7% and +10.8%, respectively, while the level of CA 15-3 in the HER2(+) and HER2(−) groups was lower at GIII +27.3% and +17.9%, respectively. CA 27.29 behaved ambiguously. In the HER2(+) group, its level was higher at GIII by +70.5%, while in the HER2(−) group, on the contrary, it was lower by −21.5%. An equally high level was shown by CA 15-3, CA 27.29, and MCA in the HER2(+) and HER2(−) groups with a high proliferative activity index.

When comparing different molecular biological subtypes of breast cancer, it was shown that a decrease in the level of MUC1 CA 15-3, CA 27.29, and MCA was observed only for HER2-positive subtypes (luminal B (HER2+) and non-luminal) (Figure 2). An increase in the level of MUC1 CA 15-3, CA 27.29, and MCA was observed only for luminal A breast cancer.Figure 2Relative change in mucin level depending on the molecular biological subtype of breast cancer compared to the control group, %. *—differences with healthy controls are statistically significant; **—differences with fibroadenomas are statistically significant; *p* < 0.05. Here and further in Figure 3 and Figure 4, relative changes are calculated as the difference between the corresponding value for breast cancer and healthy controls relative to healthy controls (Δ, %).
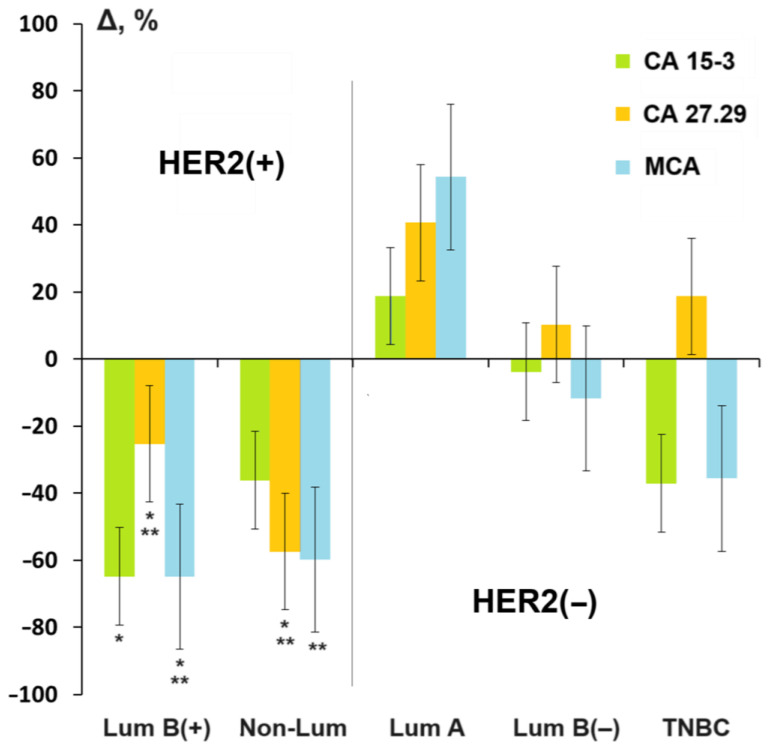


### 3.3. The Level of CA 15-3, CA 27.29, MCA in Saliva Depending on Hormonal Status

In the next step, we determined the level of progesterone and estrogen in saliva depending on the characteristics of the studied subgroups (Table 4). For the healthy control, the level of estrogen and progesterone in saliva was 3.62 [2.63; 5.20] and 2.67 [2.34; 3.08] nmol/L, respectively. A decrease in the level of free estrogen and progesterone in saliva was observed among patients with breast cancer compared to the control group (Table 4), which is consistent with the statement about the direct relationship between the level of steroid sex hormones and the level of mucin expression. It was found that the level of estrogen and progesterone in saliva statistically significantly increased with a high degree of differentiation of breast cancer cells, a high index of proliferative activity, and the presence of the expression of hormone receptors (Table 4). An increase in the level of free estrogen was observed in the HER2(+) molecular biological subtype of breast cancer, whereas no significant changes were shown for progesterone.cancers-16-03461-t004_Table 4Table 4Salivary levels of progesterone and estradiol in breast cancer, depending on the presence of ER, PR expression, the degree of cell differentiation, and the proliferative activity index Ki-67.SubgroupsProgesterone, nmol/L*p*-ValueEstradiol, nmol/L*p*-Value**HER2 status**HER2(−), n = 472.44 [1.83; 3.04]*p* = 0.49392.93 [2.55; 3.36]*p* = 0.1265HER2(+), n = 322.50 [2.26; 2.90]3.23 [2.60; 3.99]**Estrogen (ER) status**ER(−), n = 472.81 [2.42; 3.44]*p* = 0.00033.36 [2.94; 3.99]*p* = 0.0006ER(+), n = 632.27 [1.77; 2.61]2.78 [2.33; 3.23]**Progesterone (PR) status**PR(−), n = 592.78 [2.43; 3.30]*p* = 0.00043.36 [2.93; 4.03]*p* = 0.0001PR(+), n = 512.26 [1.84; 2.54]2.69 [2.33; 3.12]**Degree of differentiation (G)**GI + II, n = 462.35 [1.91; 2.66]*p* = 0.03872.73 [2.46; 3.24]*p* = 0.0015GIII, n = 472.63 [2.08; 3.41]3.36 [2.84; 4.13]**Proliferative activity index Ki-67**Ki-67 low, n = 342.34 [1.80; 2.62]*p* = 0.02382.84 [2.38; 3.26]*p* = 0.0490Ki-67 high, n = 762.60 [2.20; 3.23]3.08 [2.69; 3.93]

### 3.4. Changes in Salivary Cytokine Levels Depending on HER2 Expression in Breast Cancer

At the next stage, we compared the level of cytokines as another potential regulator of mucin expression in breast cancer depending on the HER2 status (Table 5).
cancers-16-03461-t005_Table 5Table 5Salivary cytokine content in breast cancer, depending on HER2 status, in fibroadenomas and in the controls.CytokineBreast CancerHealthy Control, n = 59 (Group 3)Fibroadenomas, n = 92 (Group 4)HER2(+), n = 62 (Group 1)HER2(−), n = 159 (Group 2)**TNF-α, pg/mL**2.56 [2.16; 3.54]2.65 [1.76; 3.85]2.43 [2.07; 3.32]2.47 [1.33; 3.02]**MCP-1, pg/mL**56.74 [32.39; 115.0]50.07 [35.43; 95.07]46.14 [35.43; 120.1]46.46 [23.75; 88.43]**VEGF, mE/mL**1338.7 [727.4; 1986.7]1270.1 [579.4; 2119.5]496.1 [352.7; 1360.2]1081.1 [505.4; 1667.4]*p1-3* = 0.0140*p2-3* = 0.0087*p1-3* = 0.0140*p2-3* = 0.0087-**IL-1β, pg/mL**122.6 [29.43; 210.1]129.8 [30.84; 318.2]37.01 [11.78; 106.1]144.5 [37.00; 310.1]*p1-3* = 0.0119*p2-3* = 0.0053*p1-3* = 0.0119*p2-3* = 0.0053*p3-4* = 0.0061*p3-4* = 0.0061**IL-2, pg/mL**6.35 [4.85; 7.85]4.04 [2.33; 9.89]1.98 [1.15; 6.71]2.43 [1.84; 5.85]*p1-2* = 0.0407*p1-3* < 0.0001*p1-4* < 0.0001*p1-2* = 0.0407*p2-3* = 0.0005*p2-4* = 0.0102*p1-3* < 0.0001*p2-3* = 0.0005*p1-4* < 0.0001*p2-4* = 0.0102**IL-4, pg/mL**3.34 [2.30; 4.97]2.32 [1.75; 3.48]1.61 [1.03; 2.96]2.79 [1.74; 4.88]*p1-2* = 0.0021*p1-3* < 0.0001*p1-2* = 0.0021*p2-3* = 0.0003*p1-3* < 0.0001*p2-3* = 0.0003*p3-4* = 0.0003*p3-4* = 0.0003**IL-6, pg/mL**3.70 [3.08; 4.99]3.19 [2.03; 5.22]4.39 [2.78; 6.63]2.78 [1.54; 5.96]-*p2-3* = 0.0496*p2-3* = 0.0496*p3-4* = 0.0453*p3-4* = 0.0453**IL-8, pg/mL**59.13 [22.61; 140.2]67.72 [24.58; 141.0]101.3 [22.73; 187.1]55.73 [21.22; 183.1]**IL-10, pg/mL**5.76 [4.00; 7.86]4.53 [3.07; 6.53]2.25 [1.68; 3.48]4.97 [3.78; 6.38]*p1-2* = 0.0099*p1-3* < 0.0001*p1-2* = 0.0099*p2-3* < 0.0001*p1-3* < 0.0001*p2-3* < 0.0001*p3-4* < 0.0001*p3-4* < 0.0001**IL-18, pg/mL**65.00 [32.27; 111.6]67.05 [30.03; 141.4]63.86 [22.50; 141.8]62.50 [40.90; 134.3]

We found that in the HER2(+) group, compared with the control, the highest level was shown by the following cytokines: MCP-1 (56.74 pg/mL), VEGF (1338.7 mE/mL, *p* = 0.0140), IL-2 (6.35 pg/mL, *p* < 0.0001), IL-4 (3.34 pg/mL, *p* < 0.0001), and IL-10 (5.76 pg/mL). At the same time, the level of IL-6 in saliva in the HER2(+) group was much lower (−15.72%) than in the control group, but higher by +15.99% compared with the HER2(−) group. The lowest level of IL-8 in saliva compared to the control was shown in the HER2(+) group (59.13 pg/mL), but it was higher than in fibroadenomas (55.73 pg/mL). In the HER2(−) group, compared to the control and fibroadenomas, the highest level was shown for cytokines TNF-α (2.65 pg/mL) and IL-18 (67.05 pg/mL). An interesting observation was that the maximum high value of IL-1β (144.5 pg/mL) was shown in fibroadenomas and the lowest value was in HER2(+) status (122.6 pg/mL). In addition, in the group with fibroadenomas, the lowest level compared to HER2(+)/(−) and control groups was shown for such cytokines as IL-6 (2.78 pg/mL), IL-8 (55.73 pg/mL), and IL-18 (62.5 pg/mL). We assume that the increased level of cytokines MCP-1, VEGF, and IL-2 to a greater extent reflects the current state of the immune system, in which the activation of typical agents of the pro-inflammatory reaction, for example, IL-1β, has already passed and is suppressed by the activity of cancer cells, which is a protective mechanism for their survival. This statement is supported by an increase in anti-inflammatory cytokines IL-4 and IL-10, both as an evasion of the immune response to tumor cell growth and due to relatively high levels of estrogen and progesterone, which also activate anti-inflammatory cytokines.

We next compared the levels of cytokines previously shown to be associated with varying HER2 expression across different molecular biology subtypes of breast cancer (Figure 3). It was shown that luminal B(+) and non-luminal subtypes, as well as luminal A and B(−) subtypes, were similar in the nature of changes, whereas the triple-negative subtype of breast cancer differed significantly.
Figure 3Salivary cytokines in different molecular biological subtypes of breast cancer (relative change in level compared to the control group, %). *—differences with healthy controls are statistically significant (*p* < 0.05).
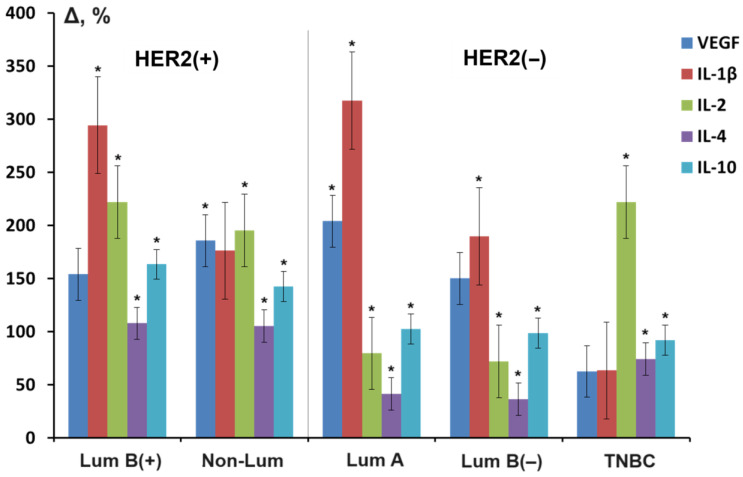


### 3.5. Changes in the Level of Free Salivary Amino Acids Depending on HER2 Expression in Breast Cancer

In the next step, we measured the level of amino acids in saliva in breast cancer, fibroadenomas, and the control group (Table 6). The purpose of measuring amino acids in saliva in breast cancer was to detect a characteristic shift in the metabolic process in general in breast cancer and depending on the molecular biological subtype. In this work, we mention only those amino acids whose level depended on the expression status of HER2 receptors (Table 6). We have shown that the level of all amino acids was statistically significantly increased in the HER2(+) group compared to the control: Asp (+65.5%, *p* = 0.0327), Gly (+60.1%, *p* = 0.0261), His (+15.9%, *p* = 0.0029), Leu + Ile (+155.2%, *p* = 0.0140), Orn (+103.0%, *p* = 0.0006), Phe (+47.2%, *p* = 0.0275), Pro (+81.1%, *p* = 0.0394), Tyr (+70.2%, *p* = 0.0231). Only Gln showed a decrease in level by 41.9% compared to the healthy control. It is important to note that in the group with fibroadenomas, a significant decrease was shown in all amino acids compared to the control group. Only the Tyr level was higher than normal values by +1.63% but lower compared to HER2(+) by −40.3% (*p* = 0.0010) and HER2(−) by −28.4% (*p* = 0.0007).cancers-16-03461-t006_Table 6Table 6The level of free salivary amino acids in breast cancer, depending on HER2 status, in fibroadenomas and in the controls, nmol//L.AAsBreast Cancer
Healthy Control, n = 25 (Group 3)Fibroadenomas, n = 24 (Group 4)HER2(+), n = 28 (Group 1)HER2(−), n = 88 (Group 2)**Asp**20.55 [17.16; 23.22]16.51 [9.30; 21.65]12.42 [7.80; 21.79]9.57 [8.20; 13.22]*p1-3* = 0.0327*p1-2* = 0.0326*p1-4* = 0.0004*p1-2* = 0.0326*p2-4* = 0.0252*p1-3* = 0.0327*p1-4* = 0.0004*p2-4* = 0.0252**Gln**255.13 [238.78; 611.84]209.79 [89.83; 394.18]438.76 [163.7; 638.4]180.6 [114.5; 439.6]-*p2-3* = 0.0338*p2-3* = 0.0338-**Gly**299.30 [218.39; 395.03]239.10 [145.98; 370.94]186.95 [141.7; 305.6]160.1 [144.8; 206.4]*p1-3* = 0.0261*p1-4* = 0.0006-*p1-3* = 0.0261*p1-4* = 0.0006**His**76.64 [62.77; 87.04]64.50 [55.98; 80.95]66.12 [55.96; 96.48]58.90 [55.42; 64.77]*p1-4* = 0.0029--*p1-4* = 0.0029**Leu + Ile**94.94 [77.38; 117.37]71.31 [23.77; 109.81]37.21 [14.17; 68.03]24.50 [15.76; 32.77]*p1-3* = 0.0140*p1-2* = 0.0237*p1-4* = 0.0001*p1-2* = 0.0237*p2-4* = 0.0084*p1-3* = 0.0140*p1-4* = 0.0001*p2-4* = 0.0084**Orn**68.27 [40.79; 95.42]45.33 [28.04; 77.58]33.63 [20.45; 46.21]25.92 [19.62; 44.20]*p1-3* = 0.0006*p1-2* = 0.0188*p1-4* = 0.0001*p1-2* = 0.0188*p2-3* = 0.0269*p2-4* = 0.0024*p1-3* = 0.0006*p2-3* = 0.0269*p1-4* = 0.0001*p2-4* = 0.0024**Phe**57.72 [52.09; 71.00]52.38 [42.04; 63.57]39.20 [30.79; 62.09]34.88 [26.79; 42.23]*p1-3* = 0.0275*p1-2* = 0.0393*p1-4* < 0.0001*p1-2* = 0.0393*p2-4* < 0.0001*p1-3* = 0.0275*p1-4* < 0.0001*p2-4* < 0.0001**Pro**135.75 [102.13; 220.13]121.64 [83.22; 172.63]74.96 [63.51; 189.41]73.52 [59.84; 112.3]*p1-3* = 0.0394*p1-4* = 0.0006*p2-4* = 0.0010*p1-3* = 0.0394*p1-4* = 0.0006*p2-4* = 0.0010**Tyr**161.43 [96.56; 205.89]134.66 [101.47; 199.42]94.85 [72.96; 170.0]96.40 [60.95; 112.9]*p1-3* = 0.0231*p1-4* = 0.0010*p2-3* = 0.0257*p2-4* = 0.0007*p1-3* = 0.0231*p2-3* = 0.0257*p1-4* = 0.0010*p2-4* = 0.0007

We noted an increase in the level of Leu + Ile in all molecular biological subtypes of breast cancer (Figure 4).Figure 4Free amino acids in saliva in different molecular biological subtypes of breast cancer (relative change in level compared to the control group, %). *—differences with healthy controls are statistically significant (*p* < 0.05).
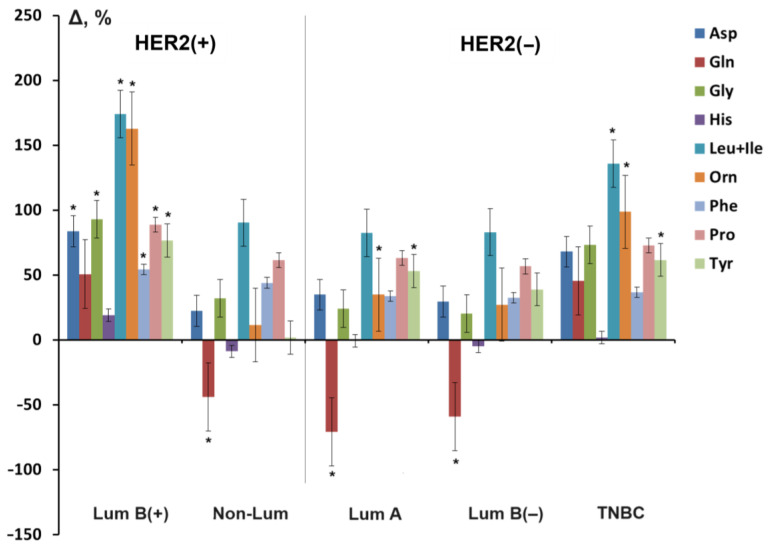


In the groups of luminal B (+) and non-luminal subtypes, which can be conditionally defined as HER2(+), the change in the level of amino acids occurred unevenly. Thus, with the luminal B (+) subtype, obviously high values were seen in Leu + Ile, Orn, Gly, Pro, Asp, Tyr, and Gln, while with the non-luminal subtype, Leu + Ile, Pro, Phe, Gly, and Asp increased with a simultaneous decrease in Gln. According to the data obtained, we see that the change in the composition of amino acids occurred approximately equally in luminal B (+) and TNBC, as well as non-luminal, luminal A, and luminal B (−), where His and Gln were reduced.

## 4. Discussion

In our study, we attempted to analyze the change in the level of salivary mucins in breast cancer and explain the molecular biological causes of changes in their level in controls, in fibroadenomas, and in breast cancer. We showed a statistically significant decrease in the content of CA 15-3, CA 27.29, and MCA compared to the control group. An increase in the level of CA 27.29 and MCA in saliva in fibroadenomas compared to the control group is noteworthy. According to literary sources, the level of mucins in the blood serum increases in breast cancer [62,63,64]. We register a reverse redistribution of the level of mucins in saliva. Hypothetically, this can be explained by the different nature of mucins circulating in different biological fluids and produced by different cells. Thus, an increase in the level of mucins in the blood serum is due to the aberrant expression of this glycoprotein, which transfers them to the status of tumor-associated mucins [20]. This leads to a change in the regulation of mucin expression and the inactivation of their anti-inflammatory functions. Normally, the variable number tandem repeat region (VNTR) of the extracellular domain of mucin MUC1 is intensively glycosylated [65,66], due to which the antigenic epitopes of the cell are hidden from the major histocompatibility complex I and mask the Toll-like receptor, which reduces the effector functions of T cells [67]. Mucin suppresses the level of cytokines, implementing their reverse regulation of cytokine production [68,69,70]. Tumor-associated mucins have a sparse, short, and altered glycosylated extracellular domain, resulting in a loss of cell polarity. Transmembrane domains of mucins approach receptors on the cell surface and the quality of its receptor apparatus changes, due to which the cell’s ability to differentiate, its proliferative activity, and the invasion and metastasis of cancer cells changes [71,72]. The cells have exposed antigenic epitopes that activate the immune system and stimulate the production of cytokines [65,73]. One study suggested that during inflammation and in the inflamed microenvironment of the tumor, mucins play an anti-inflammatory role, while aberrant tumor-associated mucins play a pro-inflammatory role [20]. Since mucins are not filtered through the hematosalivary barrier due to their large molecular weight of 300–450 kDa [74,75,76], a decrease in the levels of CA 15-3, CA 27.29, and MCA occurs at the local level in saliva. We assume that with the most aggressive molecular biological subtype of breast cancer, at widespread stages of the disease, with low cell differentiation and a high level of proliferative activity, the number of tumor-associated mucins increases, stimulating the production of cytokines. An increased amount of cytokines easily passes the hematosalivary barrier and, here, the classical type of regulation of mucin expression on physiologically unchanged normal cells is realized, in which cytokines suppress the expression of mucins. It is known that one factor may not be enough to suppress the expression of mucins in saliva. In general, immune protection in the aggressive nature of breast cancer weakens due to the redistribution of products that ensure the vital activity of cells toward cancer cells, as a result of which the functionality of normal cells, in general, and the anti-inflammatory activity of mucins, in particular, is disrupted. We assumed that the level of mucins in saliva would not depend so much on the stage of breast cancer, the degree of differentiation, or the level of proliferative activity, but on the presence or absence of HER2 expression. According to our results, the levels of CA 15-3, CA 27.29, and MCA in HER2(+) breast cancer were significantly lower than in the HER2(−) group.

Next, we assessed the mucin level in saliva depending on the presence or absence of HER2 expression relative to the stage of cancer, the degree of cell differentiation, and the level of proliferative activity. It was noted that CA 15-3 and MCA behaved in the same way regardless of HER2 expression; namely, they increased at a low stage of cell differentiation (GIII). CA 27.29 increased at GIII only in the HER2(+) subgroup, while in the HER2(−) group, an increase was registered at GI-II. Despite the fact that the MUC1 level in the HER2(+) subgroup is maximally reduced, we observed relative increases in advanced breast cancer stages, as well as in cases of low cell differentiation and high proliferative activity index. We assume that this is due to a weakening of the aggressive immune system response and a decrease in the release of pro-inflammatory cytokines that reduce MUC1 expression. The multidirectional change in CA 27.29 depending on the stage of the disease and the degree of cell differentiation relative to the presence of HER2 expression cannot be explained now and is a topic for future research.

The reasons for the differently directed changes in mucin level depending on HER2 expression and disease status are also unclear. Thus, CA 15-3 and CA 27.29 in the HER2(+) group increase at the early stages of the disease, while in the HER2(−) group, the increase occurs at advanced stages of breast cancer. Here, we can assume that HER2(+) has a more aggressive growth pattern, more aggressive suppression of the immune response, and evasion of cancer cells from cytokine activity, which causes a higher level of mucins. We can also explain the increased content of CA 15-3 and CA 27.29 at advanced stages due to the less aggressive nature of the oncological process, with greater activation of the immune response, and, as a consequence, suppression of the expression of CA 15-3 and CA 27.29. At the same time, the question of the reasons for the increase in MCA in the HER2(+) group at advanced stages of the disease remains unresolved, while in the HER2(−) group, the increase is observed at early stages.

Unidirectional change toward the increase of MUC1 CA 15-3, CA 27.29, and MCA levels was shown in the HER2(+) group with a high index of proliferative activity Ki-67 compared to the HER2(−) group. As is known, the percentage of Ki-67 activity is used for the differentiation of molecular biological subtypes of luminal A and luminal B breast cancer [77]. In addition, it is closely related to the aggressive nature of cancer course and tumor cell proliferation [78,79]. In our case, the more aggressive course of the cancer process leads to the suppression of pro-inflammatory immune response, and, consequently, to higher levels of CA 15-3, CA 27.29, and MCA.

The hypothesis about the decrease in the level of mucins in saliva due to the increased level of cytokines and decreased level of steroid sex hormones is confirmed by the results obtained in our study. We observe an increased level of cytokines, mostly pro-inflammatory in nature—VEGF, IL-1β, IL-2, and IL-18—but also an increase in anti-inflammatory cytokines IL-4 and IL-10 in the HER2(+) group compared to the control group. There are high levels of VEGF, IL-2, IL-4, and IL-10 in the HER2(+) group compared to the HER2(−) group. We also showed that the level of estrogen and progesterone is significantly lower among patients with breast cancer compared to the control. The low level of steroid sex hormones among patients with breast cancer is due to the postmenopausal period. The reasons why the level of hormones in postmenopausal breast cancer is lower than among healthy controls remain to be explained.

It is known that the functionality of the immune system is determined by humoral regulation through steroid hormones such as progesterone and estrogens [80]. A recent study has shown how elevated levels of estrogens, interacting with the membrane-bound estrogen receptor, which is coupled with G proteins (GPR30/GPER-1), regulate the activity of Toll receptor 4 on the cell surface of macrophages. Due to this, the regulation of the polarization of macrophages by the M2 type occurs, thereby performing anti-inflammatory functions [81]. Välimaa H. et al. found that the oral mucosa is sensitive to estrogen due to the presence of ERα and ERβ receptors. The ambiguous expression of ER in general by epithelial cells of the oral cavity is explained by the uneven redistribution of its two subtypes [82,83]. In addition, it has been shown that an activated ERα receptor on the cell surface of macrophages inhibits the synthesis of pro-inflammatory cytokines [84,85,86] and also suppresses the transcriptional activity of NF-kB, which is activated due to the accumulation of lipopolysaccharides during inflammatory reactions [87]. According to the literature, the regulation of the level of MUC1 is also carried out due to the expression of the steroid sex hormones estrogen and progesterone [88]. A correlation was shown between the increased expression of MUC1 mRNA and the level of estrogen and progesterone in MCF7 and ZR75-1 breast cancer cell lines [89,90]. The stimulated expression of MUC1 on the cell surface, due to increased levels of estrogen and progesterone, affects the phenotypic identity and predisposition to the invasive growth and metastasis of cancer cells [91]. It is known that estrogens and progesterone are both direct and indirect regulators of mucin production. Our study showed a reduced level of free estrogen and progesterone in saliva among patients with breast cancer compared to the control group (Table 4), which is consistent with the statement about a direct relationship between the level of steroid sex hormones and the level of mucin expression. At the same time, the relative increase in hormone levels with their absolute decrease compared to the control group characterizes the aggressive nature of tumor growth in the HER2(+) molecular biological subtype of breast cancer. We hypothesize that the low levels of MUC1 (CA 15-3, CA 27.29, and MCA) are mainly due to the patients’ decreased hormonal levels. This fact can both directly affect the expression of MUC1 and indirectly affect it through the suppression of the production of anti-inflammatory cytokines and the activation of pro-inflammatory cytokines, which, in turn, also have a suppressive effect on the expression of MUC1 in the oral cavity.

Changes in the amino acid composition in saliva reflect systemic and local changes in the metabolic activity of the body occurring in breast cancer, in particular in the HER2(+) subtype. We have shown that in the group of HER2(+) breast cancer patients, there is a statistically significant increase in the amino acids Leu + Ile, Orn, Pro, Tyr Asp, Gly, Phe, and His compared to the HER2(−) group and the controls. The level of Gln in the HER2(+) group is higher than in HER2(−) but lower compared to the values in the control group. It is advisable to consider the increase in individual amino acids in a group of biochemical processes occurring in the human body to understand the reason for their increase/decrease in breast cancer.

It is known that the HER2(+) subtype of breast cancer has an aggressive growth pattern, is highly invasive, and metastasizes. Fast-growing cancer cells constantly require fuel and building materials for their survival, which is accompanied by the formation of an inflammatory environment of the tumor and, as a consequence, the activation of the immune system occurs, which also requires building materials and energy [92,93]. Taking into account the above, we see that both the growth activity of cancer cells and the activity of the immune system are provided by amino acids such as Leu + Ile, Phe, Tyr, Asp, Gly, Gln, and Orn. It is known that high levels of Leu + Ile through the production of Acetyl-CoA, as well as Phe and Tyr through the formation of fumarate [94], feed the TCA cycle, providing cells with the necessary amount of ATP [95,96,97]. It is interesting to note that the ECM in the tumor microenvironment is an alternative source of nutrients during amino acid starvation; especially Phe and Tyr are actively consumed [98]. The amino acids Asp, Gln, and Gly provide cells with the necessary carbon and nitrogen for the synthesis of purine bases, which is necessary for cell division [99]. Phe and Tyr are also involved in the positive regulation of the cell cycle through mTOR1, thereby activating cell growth and proliferation [100,101]. An amino acid such as Orn acts as a substrate for polyamines, which promote the growth of cancer cells due to the activation of eukaryotic translation initiation factor 5A (EIF5A), which initiates protein translation [102]. Leu + Ile also make a significant contribution to the regulation of cell proliferation by positively affecting mTOR1 through the dephosphorylation of sestrin2, thereby reducing the inhibitory effect of gastor 2 on mTOR1, which leads to cell growth [103,104]. The metabolic pathway and nature of cell functioning are determined through genetic and epigenetic regulation. Thus, Leu + Ile directly regulates cell functionality by acetylation [105,106] and Gly by the methylation [107,108] of DNA and histones. Thus, we can roughly generalize the group of free amino acids Leu + Ile, Phe, Tyr, Asp, Gly, Gln, and Orn, which are responsible for cell division and functioning, from epigenetic and genetic regulation to the potential for proliferation. In our study, all amino acids conditionally assigned to this group showed a significant increase in saliva compared to the control group, except for Gln. The decrease in Gln can be explained by overspending on the performance of the antioxidant function, which, in all likelihood, is the leading one [109].

The active growth of cancer cells in HER2(+) breast cancer is accompanied by an increasing hypoxic state of tissues. We observed increased levels of free Asp and Orn in saliva with a simultaneously high level of VEGF, which indicates the activation of angiogenesis due to the increasing hypoxic state during the active division of cancer cells, as evidenced by a high index of proliferative activity [110]. Ornithine produces citrulline through the enzyme ornithine transcarbomylase (OTC) [111,112]. Aspartate interacts with L-citrulline under the influence of arginine succinate synthase to form arginine succinate, which is further broken down into L-arginine and fumarate [113]. L-arginine is a substrate for the formation of NO [114], which directly activates VEGF [115]. The increasing hypoxic state of tissues is also contributed by the increased synthesis of collagen fibers, which surround the region of tumor cells, protecting them from attack by the immune system [116]. The high content of collagen fibers is indicated by the increased value of Pro, which is an integral component of the collagen fibers [117]. We have shown that a group of amino acids that can increase in a hypoxic state, namely Asp, Orn, and Pro, have significantly high levels in saliva in the HER2(+) molecular biological subtype of breast cancer, which is consistent with information about the aggressive rapid nature of cancer cell growth in which hypoxic tissue changes are inevitable.

In our study, we observed the hyperactivity of the immune system. The trigger for this process is the active growth of cancer cells, in which the cells have a changed structure and can be perceived by the cells as foreign agents [118,119]. The active growth of cancer cells causes damage to the healthy tumor microenvironment, which is also an inflammatory trigger [120]. Increasing hypoxia, high levels of reactive oxygen species, and lipid peroxidation are also powerful activators of the immune response [121,122]. We have shown an increase in the amino acids Gly, Asp, Orn, and His, which can act as antioxidants and regulators of the immune system. Gly is necessary for the synthesis of glutathione (GSH) as the main antioxidant [123,124]. Orn also reduces oxidative stress during active protein destruction and the production of large amounts of ammonia, due to the inclusion of ammonia in the ornithine cycle and urea synthesis [125,126]. Free His, under the action of the enzyme histidine decarboxylase, is converted into histamine, which in turn binds to histamine receptors (H4R) [127] on many cells (mast cells, T-lymphocytes, dendritic cells, macrophages, eosinophils, basophils [128]. In addition, His induces the production of IL-10, performing an anti-inflammatory function [129].

The limitations of this study include the lack of data on changes in the oral microbiome in breast cancer patients, which may also have influenced the reduction in MUC1 levels. At this stage of the study, there is no answer to the question of what contributes more to the suppression of MUC1 expression: a decrease in estrogen level, an increase in the level of pro-inflammatory cytokines, or a change in the composition of the microbiome. We have also not yet assessed the diagnostic and prognostic significance of changes in the level of salivary MUC1 in breast cancer.

## 5. Conclusions

Despite the fact that today, mucins are widely used in the diagnosis of breast cancer, the reasons and mechanisms by which the expression of mucins changes and their further role in the development and maintenance of the oncological condition remain not fully understood. We have shown that the levels of CA 15-3, CA 27.29, and MCA in saliva significantly decreased among patients with the molecular biological subtype HER2(+) of breast cancer compared to HER2(−), the control group, and fibroadenomas. At the same time, we showed a statistically significant increase in the cytokines VEGF, IL-1β, IL-2, IL-4, IL-10, and IL-18 against the background of reduced hormonal levels of estrogen and progesterone. It is worth noting that the changes occurring locally in the oral cavity reflect complex biochemical shifts in the reactivity of the immune system of the entire body. This is manifested in the suppression of the anti-inflammatory activity of MUC1 due to pro-inflammatory cytokines that have passed the hematosalivary barrier and are activated by oncogenic processes occurring in breast cancer. The suppression of the anti-inflammatory activity also occurs because of a deficiency of estrogens and progesterone, inhibiting the expression of MUC1 on the epithelial cells of the oral mucosa. This cascade of biochemical reactions occurs due to the aggressive nature of the oncological process in HER2(+) breast cancer. The association of low mucin levels with aggressive processes occurring in HER2(+) breast cancer is confirmed by the greatest decrease in CA 15-3, CA 27.29, and MCA with a high proliferative activity index Ki-67 and a low degree of differentiation of cancer cells.

## Data Availability

The raw data supporting the conclusions of this article will be made available by the authors upon request.

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
