# Peer review of "Salivary Transmembrane Mucins of the MUC1 Family (CA 15-3, CA 27.29, MCA) in Breast Cancer: The Effect of Human Epidermal Growth Factor Receptor 2 (HER2)"

_cancers, 2024, doi:10.3390/cancers16203461_

Round 1
Reviewer 1 Report
Comments and Suggestions for Authors
Dear authors:
I congratulate you in advance for this interesting work, which is of great importance in the validation of new biomarkers or combinations of markers. Your contribution to the diagnosis, treatment, etc. is highly valued. However, there are some small issues that need to be clarified for a better understanding by the readers:
1.-In lines 109 and 110, the classification between MUC and tumor antigens is unclear.
2.-It is also suggested to provide more information regarding the usefulness of these markers, especially in saliva, as of the date published.
3.-The criteria for choosing groups with different sample sizes and different ages of patients are also unclear, especially when age-related changes are mentioned in the introduction, lines 90 to 93. Please provide more clarity.
4.-The variability of age in the analyses between benign (fibroadenomas) and malignant pathologies need to be discussed, as this leaves the relevance of the statistical analysis and conclusions unclear.
5.-The graphs would be easier to understand if they started with the data from controls (HC), then fibroadenomas (FA) and finally breast cancer (BC).
6.-The legend of Figure 1 requires a little more detail and clarification of the statistical analysis and its results.
7.-In Table 3, the differences between the p-values the authors want to report are unclear and barely observable. What is the difference between p1HC, p2HC, and p1-3?
Do the values in parentheses represent the extremes, the lowest and the highest?
8.-Since the idea that Figure 2 is different from Figure 1 is intended to convey this, I suggest changing the colors of the graph to more clearly communicate that this is not HC, FA, and BC but rather different levels of MUC 12, 20, and 22. Place the name of the associated biomarker, ex MUC 20 (CA 27.29), in parentheses for better follow-up by readers who are less expert on the subject. Please indicate in the legend what statistical analysis was applied to arrive at these results and their conclusions.
9.-In Figure 3 y 4, could you explain the axes, mainly the Y axis? Also, in the legend, briefly describe the "n" analyzed for each cytokine or aminoacids according to molecular classification, the statistical analysis, and whether the graph corresponds to standard error or deviation.
10.- Although the discussion supports the hypothesis that low levels of hormones such as estrogen and progesterone, combined with higher levels of inflammatory cytokines, it does not speculate much about the mechanism of modulation of the different MUCs and leaves some questions open: Are all the MUCs groups modulated at different stages of the disease? How do we avoid the age effect?
Reviewer 2 Report
Comments and Suggestions for Authors
This article uses saliva samples to confirm that in HER2+ patients, mucin concentration is reduced, free estrogen and progesterone levels are reduced, and pro-inflammatory cytokines and free amino acids are increased, which is related to the aggressiveness of HER2-positive breast cancer subtypes. Through human experiments, the author aimed to confirm the correlation between saliva detection of CA 15-3, CA 27-9, and MCA performance with the breast cancer patient group. However, I still have some questions regarding this article. The question is as follows:
1. Tumor markers are typically detected using serum/plasma, but this article uses saliva specimens. Is there any relevant data to confirm that the serum, plasma, and saliva specimen data are consistent? However, this article utilizes saliva specimens rather than blood specimens. Therefore, the consistency of the two data sets cannot be confirmed, which diminishes the reliability of saliva testing.
2. There are several omissions in the Materials and Methods chapters.
(1) It is not specified whether individuals with healthy and fibroadenoma subjects also undergo the same processes as those with breast cancer. Please add. Please provide a clear list of the number of patients admitted to the control group, fibroadenomas, and breast cancer. In addition, please explain the meanings of mucins, cytokines, and amino acids, as presented in Table 1. Are the testing items simply different from one another? Why is the number of reported cases inconsistent? Please refresh this form.
(2) Many kit catalog numbers are missing, especially lines 145-149, 151-154, and 156-159.
3. The title is "Salivary transmembrane mucins of the MUC1 family (CA 15-3, CA 27.29, MCA) in breast cancer: The effect of epidermal growth factor expression". However, the abstract "shows for the first time that salivary concentrations of the MUC1 family, including MUC12 (CA 15-3), MUC20 (CA 27.29), and MUC22 (MCA), are significantly dependent on epidermal growth factor receptor (HER2) expression in breast cancer (Lines 9-10)." Please confirm the accuracy of the title. EGF? or HER2?
4. The standardization of saliva testing data and the definition of a cut-off value are under consideration. In addition, saliva testing may yield false positives and false negatives.
Minor
Please label the y-axis and indicate the number of subjects in each group in the figure. The original data will be displayed using a dot plot.
Reviewer 3 Report
Comments and Suggestions for Authors
Reviewer Comments
The article, “Salivary transmembrane mucins of the MUC1 family (CA 15-3, 2 CA 27.29, MCA) in breast cancer: The effect of epidermal 3 growth factor expression” by Dyachenko et al., reports a relationship between the expression of MUC1 family members and breast cancer status. However, the concentrations of MUC1 family members were insignificantly different between cancer patients and healthy control. Overall, the data did not support authors conclusion. Authors should carefully reanalyze all the data before publication.
The manuscript is relatively well organized and performed. However, the results should be improved as follows:
Major comments
1. There is no significant difference in the concentration of MUC12/20/22 in between breast cancer and healthy control.
2. In the introduction, authors should provide more structured information on the MUC1 family member. How many family members? How they are grouped?, etc.
3. What is the rationale for selecting MUC12, MUC20, and MUC22 among MUC1 family members?
4. In Table 1, why the n numbers are different in the same group? For example, In the Control group, n = 30 (Mucins), n = 59 (Cytokines), n = 25 (Amino acids).
5. In lines 189 – 190, where is the data for the total mucin content?
Minor comments
1. Minor English editing is required.
2. In lines 108 – 109, approved for what?
<The End>
Comments on the Quality of English LanguageMinor editing is required.
Reviewer 4 Report
Comments and Suggestions for Authors
In this paper the authors described the characterization of the levels of MUC1 family proteins in the saliva of breast cancer patient compared to normal and fibroadenomas patients. They found a correlation between positive Her2 cancers and decreased levels of MUC1 and concomitant increase in several cytokines. The study is carried out with rigorous methods but, to the general reader, it could be improved if the tables could be converted into graphs, more easy to follow and understand .
I have two concerns that authors may clarify:
1) why the analysis was done only on membrane associated MUC1 family and not on soluble mucins
2) in the methods they describe the collection of salivar samples that are treated by centrifugation at 10000 x g followed by recovery of the soluble material, so suggesting that cells have been eliminated from the sample
Is the MUC1 fraction they measure a product of tissue degradation and this degradation if any occurs in the saliva itself offer reflects the transport of mucins in the blood and the following transfer through the salivar barrier?
Although interesting, the results do not easily suggest how to use saliva as a main fluid to make diagnostics due to the complexity of concomitant events that lead to limit a decrease of MUC1 presence that could be due to many factors. A scheme in this line of reasoning would greatly help the readers.
Round 2
Reviewer 2 Report
Comments and Suggestions for Authors
The author responded appropriately to the comments raised.
Author Response
We once again express our gratitude for your careful attention to the manuscript and valuable comments.
Reviewer 3 Report
Comments and Suggestions for Authors
Reviewer Comments
The article, “Salivary transmembrane mucins of the MUC1 family (CA 15-3, 2 CA 27.29, MCA) in breast cancer: The effect of epidermal 3 growth factor expression” by Dyachenko et al., was resubmitted by authors with modification according to the reviewers suggestions. In general, the manuscript has been improved. However, the manuscript could be improved further. Please find suggestions as follows:
Major comments
1. Careful English editing by native speaker(s) is required. For example, lines 115 – 117 and lines 203 – 206.
2. Authors have changed the names of mucin 1 family members as CA 15-3, CA27.29, and MCA. In lines 40 – 46, there is no description on CA 15-3, CA27.29, and MCA. And CA 15-3, CA27.29, and MCA suddenly appeared in lines 77 – 79. Unfortunately, these names cannot be found in uniprot database. Please use official protein names and put the generic names in parentheses once (as in previous version of the manuscript). In my opinion, these may cause confusion or misunderstanding of audience.
3. In the 2.1. Study design, there are still discrepancies between the numbers described in the sentence and in the tables. For example, the number of patients is 230 in line 127; mucins and amino acids were determined 165 samples in line 135; mucins 110, cytokines 230, amino acids 116 for breast cancer in the Table 1. In addition, I cannot get the meaning of 166 and 165 samples in line 135.
4. What are the meanings of K226, K227, and K228 in line 159 and K208 and K207 in line 165?
5. What are the meanings of A-8756, A-8782, … in lines 170 – 172?
6. What are the meanings of [0.103; 0.561] and [0.434; 0.727] in lines 203 – 205?
7. In Fig 1B, authors should put the * for p = 0.0318. In addition, authors should put the number of samples for each graph.
8. What are the meanings of the number in brackets in Tables 2 – 6. Authors may present these data by dot plot distributions with mean ± SEM.
9. In Figures 2 – 3, authors should put the HER2 status for audience’s convenience. In addition, please use same categories in Tables 2 – 6 and Figures 2 – 3. Currently, HER2 (+), HER2 (-), Healthy control, Fibroadenomas are used in Tables, while Lum B(+), Non-Lum, Lum A, Lum B(-), and TNBC are used in Figures.
<The End>
Comments on the Quality of English LanguageCareful English editing by native speaker(s) is required.
Reviewer 4 Report
Comments and Suggestions for Authors
The authors gave satisfactory replies, but I would still suggest to clarify somewhere in the text that they are measuring degradation MUC products.
Round 3
Reviewer 3 Report
Comments and Suggestions for Authors
The manuscript has been improved according to reviewer's suggestion.